# LoRa-Based Low-Cost Nanosatellite for Emerging Communication Networks in Complex Scenarios

Raúl Parada [1,*], Victor Monzon Baeza [2], David N. Barraca-Ibort [3] and Carlos Monzo [3]

1 Centre Tecnològic de Telecomunicacions de Catalunya (CTTC/CERCA), Av. Carl Friedrich Gauss 7, 08860 Castelldefels, Spain
2 SIGCOM Group, Interdisciplinary Centre for Security Reliability and Trust (SnT), University of Luxembourg, 1855 Luxembourg, Luxembourg; victor.monzon@uni.lu
3 Department of Computer Science, Multimedia and Telecommunications, Universitat Oberta de Catalunya, 08018 Barcelona, Spain; dbarraca@uoc.edu (D.N.B.-I.); cmonzo@uoc.edu (C.M.)
* Correspondence: rparada@cttc.es

**Abstract:** Wireless broadband coverage has reached 95% worldwide. However, its trend is expected to stay the same in the following years, presenting challenges for scenarios such as remote villages and their surrounding environments. Inaccessibility to these areas for installing terrestrial base stations is the main challenge to bridge the connectivity gap. In addition, there are emergencies, for instance, earthquakes or war areas, that require a fast communication reaction by developing networks that are less susceptible to disruption. Therefore, we propose a low-cost, green-based nanosatellite system to provide complete coverage in hard-to-reach areas using long-range communication. The system comprises a pilot station, a base station, and a CubeSat with sensor data collector capabilities acting as a repeater. Our system can be built within hours with a 3D printer using common material, providing a flexible environment where components can be replaced freely according to user requirements, such as sensors and communication protocols. The experiments are performed in Spain by two test sets validating the communication among all components, with RSSI values below −148 dBm and the longest distance above 14 km. We highlight the reduction in the environmental impact of this proposal using a balloon-based launch platform that contributes to sustainable development.

**Keywords:** cubesat; SatCom; green communications; 3D-printed satellite

## 1. Introduction

Nowadays, wireless broadband coverage has reached 95% worldwide. In the case of remote areas like villages and surroundings, physical accessibility becomes a challenge for coverage extension. Therefore, installing a base station is, on the one hand, unfeasible and, on the other, not cost-efficient. Moreover, other than natural disasters such as earthquakes, volcanic eruptions, landslides, tsunamis, and other hazardous events, climate disasters are breaking records worldwide, according to Greenpeace [1]. In addition, human participation in wars, infrastructure damage, and mobility accidents generate emergency situations where accessibility is difficult (e.g., remote villages), and a fast reaction is required (i.e., the result of a natural disaster). Then, in those cases, it might be necessary for a system to communicate with entities to exchange vital information (e.g., emergency entities). One solution to this problem is mobile telephony via satellite, based on deploying numerous orbiting communication satellites in low orbit around the planet, providing telephone coverage practically anywhere. The main advantage, thanks to the innate nature of satellite communications (SatCom) [2], is that this allows uniform coverage due to the direct vision that is possessed in the earth–space link and is stable due to its robustness in the face of terrestrial problems. Among the drawbacks are that it does not work indoors, and above all, the price of the terminals. This work develops a nanosatellite prototype, under the CubeSat standard, of a communication system based on long-range (LoRa) technology for access to

communication in remote areas, including a Global Positioning System (GPS) tracker. A multipoint connection is established between the mobile stations. The CubeSat acts as a "repeater", obtaining data from a mobile station in an inaccessible area without coverage and sending the data to the base station. The advantage of this CubeSat is its model availability for 3D printing at anytime, anyywhere. Our system allows communication between two far-away entities using a nanosatellite as a repeater. Figure 1 displays a realistic and critical situation in which our system could be deployed.

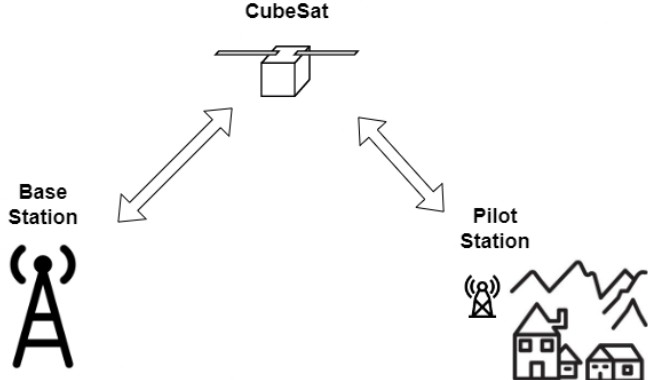

**Figure 1.** Proposed communication scheme.

This is an example of a typical scenario where a remote and isolated village might face a catastrophic event where a fast reaction is vital (e.g., a volcano eruption). The fundamental objectives of the project are to implement a communication system where a connection is established between the pilot station located in a remote place without coverage and the base station through the CubeSat, and to investigate and experiment on the possibilities of communication in remote sites such as villages or disaster areas. The communication system is equipped with LoRa technology, meaning no mobile network is necessary.

The remainder of this paper is organized as follows. Section 2 summarizes similar work projects from the literature. The design and manufacturing of the CubeSat, base station, and pilot station are detailed in Section 3. Section 4 explains the development results. The experimental evaluation is described in Section 5. Finally, the paper is concluded in Section 6, summarizing our contributions and stating future work tasks.

## 2. State of the Art

In this section, an analysis of proposals related to LoRa applied to nanosatellites is carried out, and the different communication options available for emergency scenarios are reviewed, with the aim of determining the existing gaps in the literature.

CubeSat standard is becoming more and more attractive for the emerging space industry because of its low design-and-deployment cost, which makes possible a wide range of space-related applications, such as earth remote sensing, space exploration, and rural connectivity. Further, CubeSats provide a complimentary connectivity solution to the pervasive Internet of Things (IoT) networks, leading to a globally connected cyber-physical system. In [2], a comprehensive overview of various aspects of CubeSat missions is provided, i.e., a thorough review of the topic from both academic and industrial perspectives.

An essential part of a CubeSat is the incorporation of low-powered electronics, which has also enabled further development of internet-connected devices for SatCom. Open-source platforms like Raspberry Pi and Arduino provide commercially available off-the-shelf (COTS) systems. COTS systems allow researchers to build functional prototypes faster. An example of a CubeSat system made from COTS systems is shown in [3] for a cube-shaped 1-U standard CubeSat.

Technologies like 6LoWPAN, LoRa, Sigfox, Global System for Mobile communication (GSM), Universal Mobile Telecommunications System (UMTS), and Long Term Evolution (LTE) are some standard IoT communication protocols widely used. CubeSat systems are

low-cost and consume low power. This system is being researched for IoT applications from space [3].

Among the technologies mentioned, LoRa technology has gained interest in SatCom; in [4], the authors evaluated its limitations and took it into space, demonstrating that it is becoming one of the most promising technologies for satellite IoT, particularly those based on satellite constellations in low earth orbit (LEO) [4], where the distance may be inconvenient for LoRa. Another work proposed IoT using LoRa communications for terrestrial usage [5].

In recent years, the world record for the maximum distance that a long-range wide-area network (LoRaWAN) data packet can travel has been broken several times—the last record, being 766 km, was broken by a new one of 832 km. These distances make it feasible to consider LEO systems. This experiment was made by Thomas Telkamp, CTO and co-founder of Lacuna Space, and was presented live during The Things Virtual Conference [6] in 2022, through the launch of a high-altitude balloon filled with helium, which had been connected with a LoRaWAN sensor.

High-Altitude Platform Stations (HAPS) have gained much interest recently to integrate LoRa technology for shorter distances than LEO. A LoRa transceiver via HAPS in remote areas was proposed for environmental monitoring applications in [7]. The data rate achieved by LoRa-HAPS ranges from 0.3 kbit/s to 50 kbit/s, depending on the spreading factor (SF) value. In [5], SF is 12 for a distance of 330 meters. The height of the HAPS is 10 km. An embedded HAPS with LoRa in [8] was proposed for weather balloons for this same distance. The payload for these proposals weighs 292 g and offers an output power of 300 mW. Such solutions are emerging as key proposals to improve agriculture in these underserved areas by linking them to the core network through HAPS [9]. Another solution to link areas with limited coverage to existing communication networks using balloon-based communication relays was proposed in [10], limited to a distance of 70 m. The problems with [7,8,10] are the high cost and long time for development, deployment, and launch. From the signal processing point of view, we can reduce the size of the payload by simplifying the processor by eliminating the overhead created by the transmission of pilot signals by making non-coherent schemes, as in [11].

Simplifying the satellite structure for nanosatellites allows for the design and manufacture of a functional satellite at a low cost. In addition, the encapsulation and interface of the payload reduce bureaucratic problems and prohibitions between the launcher and the developer due to CubeSat standard specifications. This is the case with picosatellites, such as those manufactured by Fossa Systems, an association whose objective is to develop and launch orbiting picosatellites, expanding access to aerospace technology and reducing its development and launch cost. Fossa Systems began the development of the first Spanish picosatellite, FOSSASAT-1, in 2018. It is a PocketQube (1P) picosatellite [12], measuring $5 \times 5 \times 5$ cm, weighing 250 g, and open-source, which is experimenting with spread spectrum technologies for the IoT.

In addition to the distance, another drawback of the LEO constellation, from a communications point of view, is the Doppler effect [13]. Laboratory testing and outdoor experiments were conducted to explore the Doppler effect in [14] to determine the feasibility of the LoRa modulation in CubeSat–SatCom systems. The experiments showed that the LoRa modulation has very high immunity to the Doppler effect, which allows for the use of LoRa in orbits above 550 km without any restrictions associated with the Doppler effect. In lower orbits, the dynamic Doppler effect leads to the destruction of the satellite-to-Earth radio channel using the LoRa modulation mode with a maximum SF of 12, as shown in [15]. This destruction occurs when the satellite is flying directly above the ground station, reducing the radio communication session duration. The reduction in the duration of a communication session increases with decreasing orbit altitude, and reaches about 1 minute in an ultra-low orbit of 200 km.

Another essential element to consider in constructing a CubeSat is the antenna. For LoRa communication, a compact antenna is required to support the LoRa protocol on the

CubeSat [16]. The microstrip antenna is one of several antennas for CubeSats that are easy to fabricate and small, as shown in [16]. This one can easily be placed on various surfaces, making it relatively easy to install on the CubeSat.

As the space-to-earth channel is different from the conventional earth-to-earth one, it is vital to assess the capabilities of LoRa in this new environment. Reference [17] presented a study of different LoRa device configurations to identify the constraints for each one and determine which one is better for particular mission requirements. Also, the effect of ionospheric scintillation is assessed with an SDR-based test set-up that evaluates the performance of this technology against Humprey's ionospheric scintillation model. The obtained metric in [17] gives the received power, such as received signal strength indicator (RSSI), as a function of the intensity scintillation index and shows the robustness of the LoRa modulation in these new environments.

*Our Contribution*

Against this background, several gaps have been identified. The examples shown in [7,8,10,14,16] require several years of development and have a high cost of production, making them not suitable for emergency situations in which a rapid response is required. Therefore, in this work, we develop a prototype that requires much less design, development, implementation, and launch time than those existing in the literature. This design meets the CubeSat standard but is based on a HAPS deployment environment, ensuring the best of both systems.

Another characteristic of the prototype, presented against the state of the art, is the use of COTS in a customizable way, not only the communications part, as in [3], but also the sensory part, as COTS elements easily and is quickly adaptable to the type of mission or emergency compared to the model presented in [6]. Regarding the communication protocol, we have chosen LoRa as an example because it is the most prevalent among those analyzed in the literature, as shown in [4]. In addition, among the characteristics of the proposed CubeSat, we highlight the ease of homemade manufacturing using 3D printing compared to the existing ones, reducing size, weight, and cost, for example, compared to [7,12,16]. Another key factor compared to the alternatives in the literature is the sustainable and green nature of the prototype presented. The environmental impact is negligible since the deployment characteristics of our proposal allow the recovery of the CubeSat, which does not generate space debris and is useful for another launch.

Specifically, we achieved the following contributions:

- Design and manufacturing of an autonomous CubeSat, base station and pilot station;
- Election and measurement of electronic devices for the efficient performance of the system;
- Design, simulation, and test of a dipole antenna;
- Management of legal rights and checking the weather forecast to launch the CubeSat locally;
- Collection of sensor data and their visualization in a web-based environment;
- Performance of LoRa communication experiments in an open environment.

## 3. System Description

This section aims to explain how the system architecture components were designed and built. The complete system proposal comprises a CubeSat, a base station, and a pilot station, the last one representing a device located in a remote location. Figure 2 presents the prototype, identifying where these components are in the system architecture. In addition, the software used for all components is considered the fourth part of the proposed system.

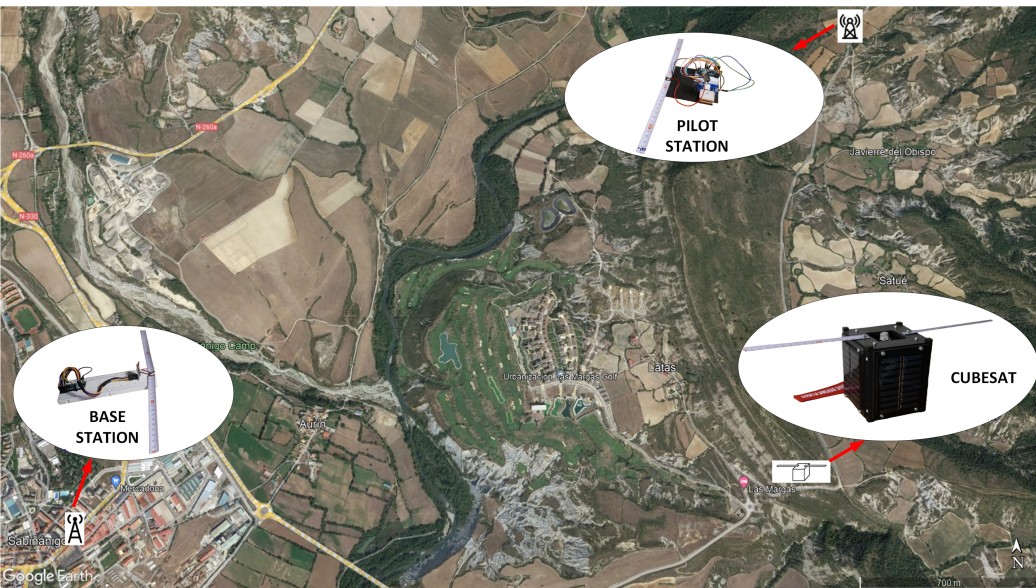

**Figure 2.** System architecture composed of pilot station, CubeSat, and base station (from left to right).

We must consider LoRa modules and antennas for communications as essential system parts. On the one hand, the chosen LoRa module for each system component is the SX1278 model from SEMTECH [18], chosen due to its long-range operation between 137 MHz and 525 MHz, spread spectrum communication capabilities, and high-frequency immunity, minimizing power consumption. On the other hand, considering the complete system development, Figure 2 shows the three components: CubeSat, base station, and pilot stations, along with a dipole antenna that was proposed to be used. With regard to antenna design, firstly, it can be manufactured simply by using, for example, a regular metallic ruler. Secondly, considering the working frequency of 433 MHz for LoRa, the antenna design establishes its width at 1.2 cm. Finally, to analyze the radiation pattern during antenna design, it was simulated using Matlab. From those simulations, we obtained the following outputs: the dipole antenna created in Matlab with its measurements (see Figure 3), the impedance chart (see Figure 4), the reflection coefficient (see Figure 5), the current distribution within the dipole (see Figure 6), the radiation pattern (see Figure 7), and diagrams of azimuth and elevation (see Figure 8). Given these, the final parameters for the proposed dipole antenna are (i) antenna width of 1.2 cm, (ii) antenna length of 34.642 cm for optimal efficiency, (iii) input impedance of 73.1 Ω, and (iv) directivity of 2.17 dBi.

Next, the main characteristics of the three components and their software are presented in Sections 3.1–3.4, describing the CubeSat, base station, pilot station, and software components, respectively.

### 3.1. CubeSat

The CubeSat component is the non-terrestrial element, which was built using a common 3D printer due to its worldwide availability. We have used the Universal 1-U CubeSat model from UltiMaker Thingiverse community [19]. This CubeSat follows a standard format of 1-U, with different levels to allocate diverse hardware components such as an Arduino Nano [20]; the estimated printing time was about 90 min. A rechargeable battery system with 8 solar panels connected in a 2 × 4 arrangement makes the CubeSat autonomous, with an energy generation of about 400 mA. As was presented previously, a dipole antenna was designed and manufactured. Considering its functionality for different applications, the proposed CubeSat includes, taking into account possible future needs, the following sensorization capabilities:

- Environmental sensor (BME280 [21]);
- Nine-axis Inertial Measurement Unit (MPU-9250 [22]);

- Air quality sensor (MQ-135 [23]);
- Ultraviolet light sensor (GUVA-S12SD [24]).

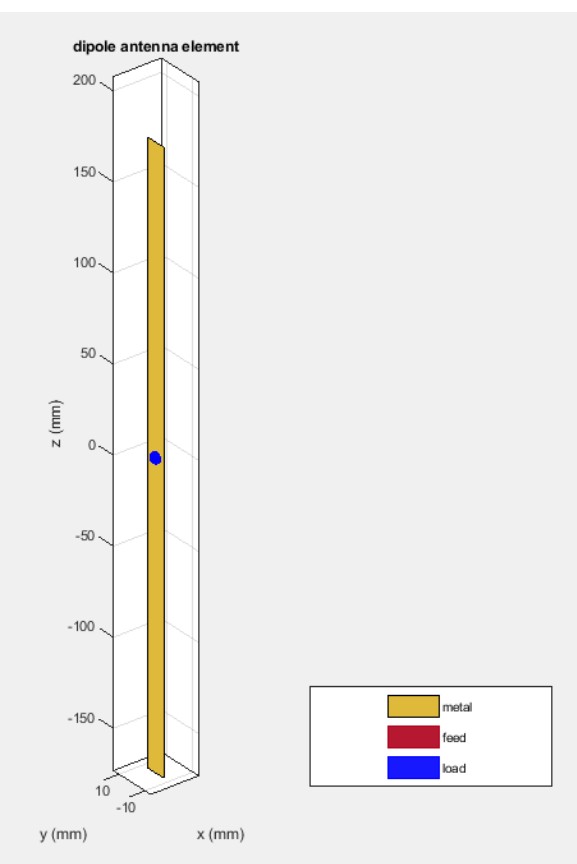

**Figure 3.** Dipole antenna on Matlab.

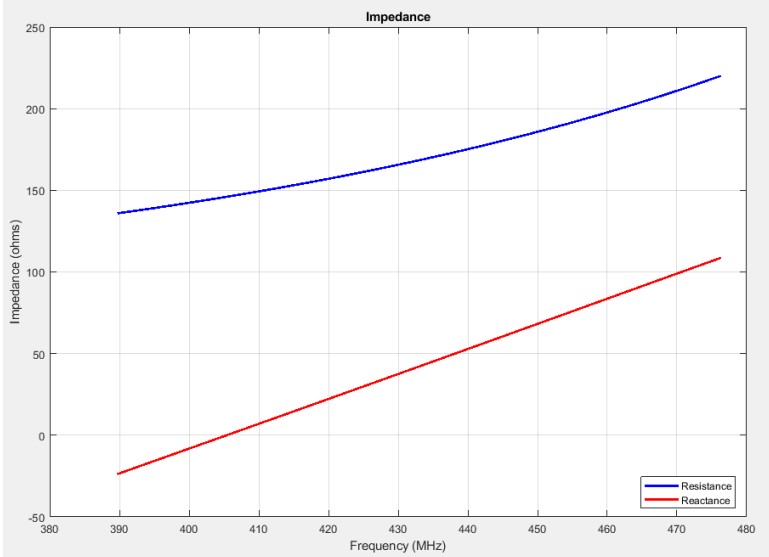

**Figure 4.** The impedance chart.

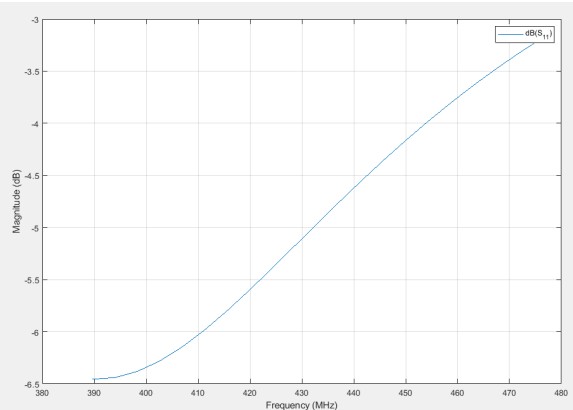

**Figure 5.** The reflection coefficient of the dipole.

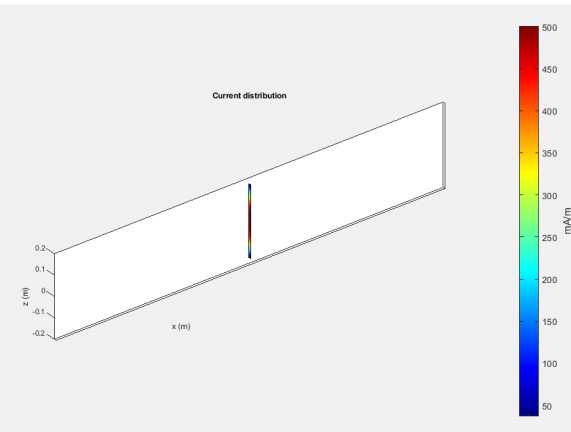

**Figure 6.** The current distribution.

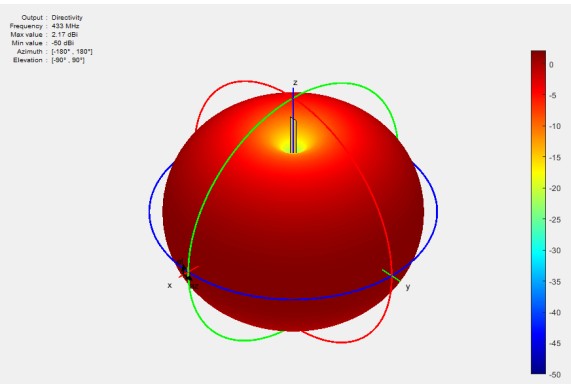

**Figure 7.** Dipole directivity.

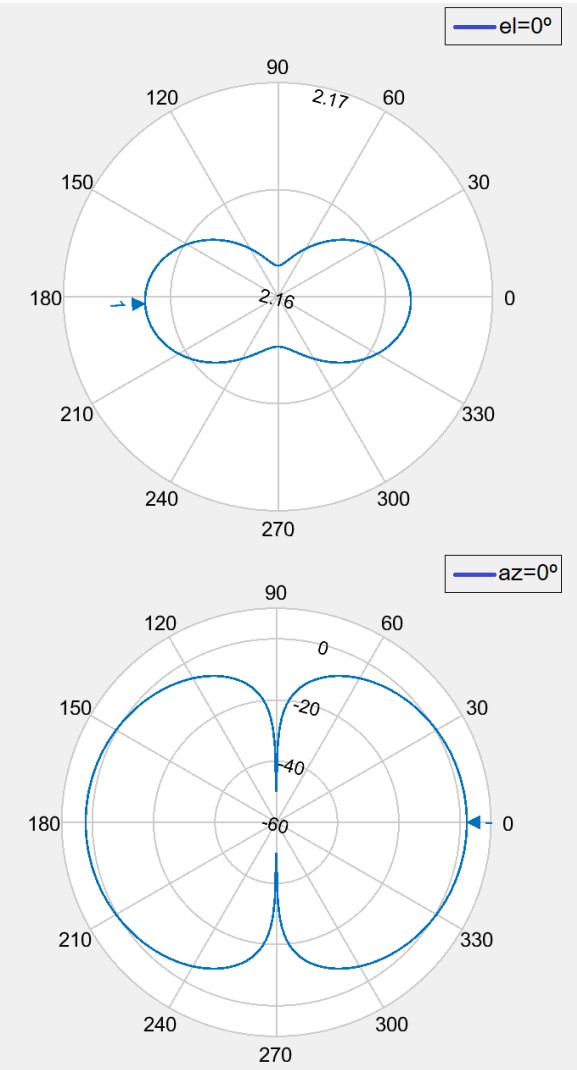

**Figure 8.** Dipole radiation pattern. The top image shows the azimuth diagram and the bottom image shows the elevation diagram.

We measured the energy consumption and summarized a total of 308.5 mA from the following elements: Arduino Nano (19 mA), LoRa SX1278 (120 mA), BME280 Sensor (1 mA), MPU-9250 Sensor (3.5 mA), MQ-135 Sensor (150 mA), GUVA-S12SD (5 mA), and other elements (10 mA). Hence, the solar panel can supply the total energy consumed by all the elements. Note that the MQ-135 Sensor consumes most of the battery. Hence, in the future, we will study less battery-depleting sensors. Note that all components are thermally protected to ensure their good performance throughout the launch. Since our solution allows a flexible configuration of components, the user might use other suitable components for her/his experimental purpose. Regarding Arduino Nano use, it performs the following tasks:

- Sensor reading: Receives the values of the analog and digital inputs of the microcontroller to process them. Taking into account the selected microcontroller and its capabilities, the reading process for different kinds of sensors is supported.
- Communication: Responsible for receiving and sending communication data between the CubeSat and the terrestrial stations through LoRa modules.

As far as power consumption is concerned for all sensors, LoRa modules, and Arduino Nano, energy needs are supplied by the solar panels.

In addition, an Arduino IDE and a Node-RED programming environment were developed to acquire and visualize sensor information, respectively. For data visualization,

a web-based dashboard was developed using Node-RED, allowing interaction with the Arduino environment. Figure 9 displays the final developed version of the CubeSat, with a total weight of 523 g.

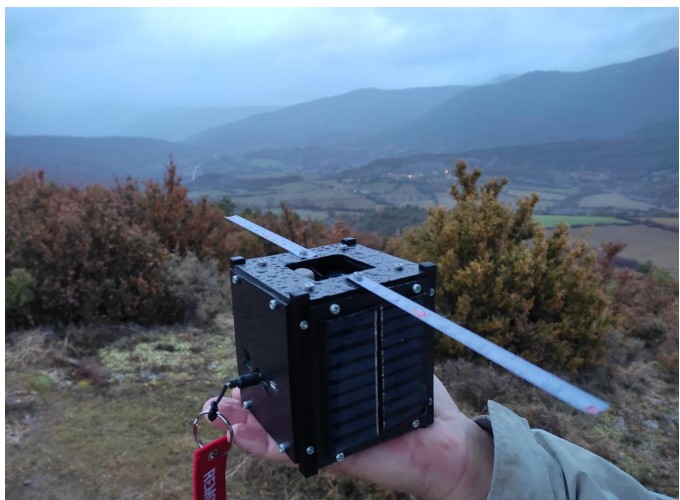

**Figure 9.** Developed CubeSat unit.

*3.2. Base Station*

The base station is the first terrestrial element to establish full LoRa communication with the pilot station, using the CubeSat as a repeater. The main difference between base and pilot stations is the location of the terminal, due to the pilot station being in a remote area and the base station being in a well-populated location. The base station is composed of a LoRa module. The antenna design followed the same dimensions and construction procedure as the one performed for the CubeSat.

In Figure 10, the electrical scheme from the base station shows how the LoRa module is connected to the Arduino Nano microcontroller.

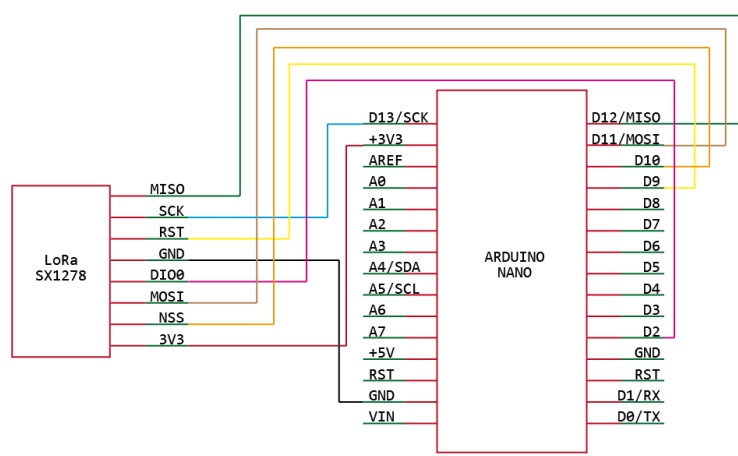

**Figure 10.** Electrical scheme for the base station.

*3.3. Pilot Station*

The pilot station is the second terrestrial component of the system, the base station being the first. Taking this into account, a flexible range of operation communication can be created between both components by using the CubeSat as a repeater. The pilot station was composed of a LoRa and a GPS module. The antenna design followed the same construction as the one developed above. Figure 11 presents how both the LoRa and GPS module interact with the microcontroller.

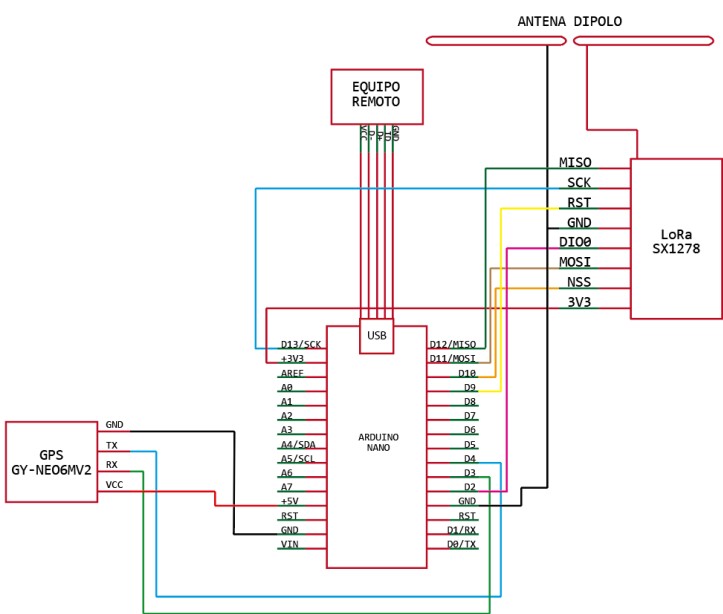

**Figure 11.** Electrical scheme for the pilot station.

Table 1 shows a summary of the total energy consumption of the pilot station located in a remote area, measured in milliamperes [mA].

**Table 1.** Pilot station consumption.

| Component | Consumption (mA) |
| --- | --- |
| Arduino Nano | 19 |
| LoRa SX1278 | 120 |
| Sensor GPS GY-NEO6MV2 [25] | 45 |
| Others | 10 |
| Total | 194 |

### 3.4. Software Components

The elements of the system were developed in Arduino using the Wiring language. The libraries provided by IDE Standard that were used in this work are the following:

- SPI.h: To use the SPI port on the Arduino to control the onboard hardware of SPI (bus communication with Arduino);
- Wire.h: To use the I2C bus in Arduino;
- LoRa.h: To send and receive data through the LoRa protocol. We used the default LoRa configuration: bandwidth = 125.0 kHz, spreading factor = 9, coding rate = 7, and output power = 17 dBm;
- TinyGPS: To provide GPS NMEA functionality;
- SoftwareSerial.h: To allow serial communication through other digital pins of the Arduino, using software to replicate the same functionality. Multiple software serial ports, with speeds up to 115,200 bps, were used.

Taking into account the previous subsections, all three elements or devices had a LoRa module to allow bidirectional communication, with the CubeSat acting as a repeater, forwarding the messages from one side to another. In terms of the electrical circuit scheme for the solution, Figure 12 details the design of the CubeSat, in which all mentioned hardware is displayed. Note that the solar panels' connection is also included for harvesting generated energy, providing power autonomy, a necessary consideration in *green environments*.

Both the base and pilot station were composed of an Arduino Nano microcontroller with a connected LoRa module connected. The base station, additionally, had a GPS module receiving position data. As observed in Figure 2, all three elements included the designed and built dipole antenna for radiocommunication. The CubeSat (nanosatellite), the base station, and the pilot station all used the SPI.h and LoRa.h libraries. In addition, the pilot station used TinyGPS and SoftwareSerial.

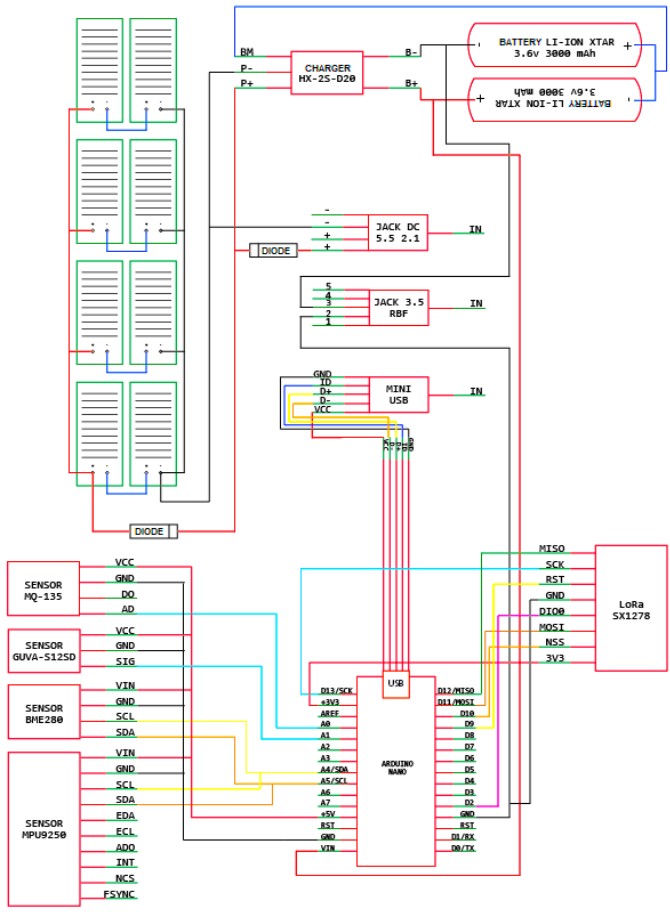

**Figure 12.** Electrical scheme of the CubeSat.

## 4. Pre-Launch Procedures

The implementation and development of this nanosatellite was performed in Spain. Notably, for future applications, this procedure must be checked according to the specific country's air regulations. The steps to successfully launch the proposed CubeSat in Spain are outlined below:

1.  Airspace reservation:
    - Choose a launch site, avoiding the existence of overhead lines nearby and reducing the likelihood of wind blowing the probe out to sea;
    - Contact the corresponding area navigation manager in the launching country to request the form—in the proposed case, ENAIRE (air navigation manager of Spain);

2.  Selection of components for deployment:
    - Choose balloon (Totex TA-1000), helium load (3.5 cm$^3$ volume), and parachute (Rocketman 4 ft);
    - Insert information into *Parachute Descent Rate Calculator* at parachute section and *Meteorological Balloon Burst Estimator* according to [26], to estimate descent rate and balloon burst, respectively;

3.  Simulation of the possible trajectories of the balloon:
    - Choose the location and time of launch;
    - Predict the balloon trajectory [27];

We used the online software in [28] to simulate the possible trajectories of the balloon for the last point. The simulated trajectory is shown in Figure 13 for the following launch data: an altitude of 874 km, date and time of 5 August 2023 at 08:00, ascent rate of 5 m/s, and burst altitude of 30,000 m. The latitude and longitude for launch were 42.539964 and −0.376872, respectively.

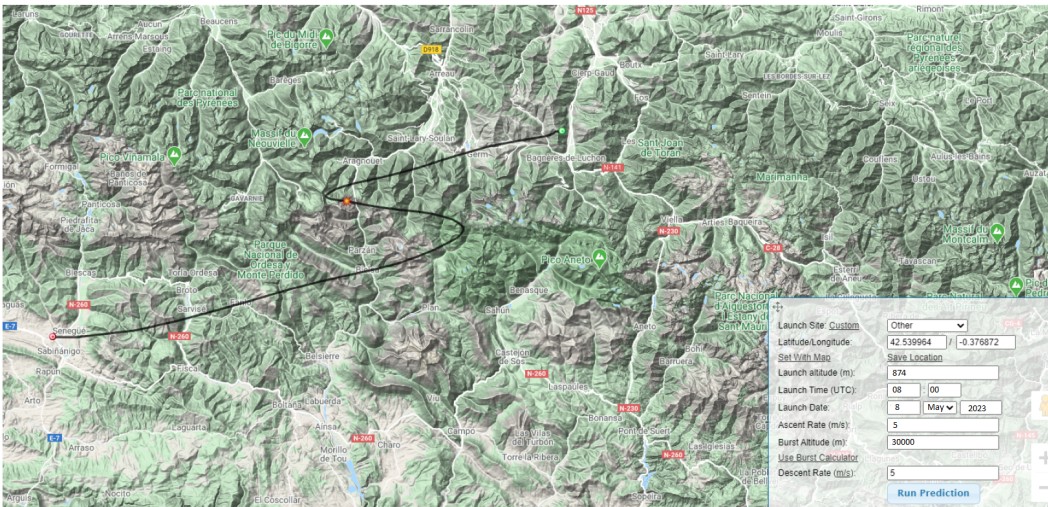

**Figure 13.** Estimation of the trajectory of the balloon.

## 5. Test Verification and Evaluation

This section shows the results of the outdoor experiments performed in a village in the Aragon region in Spain (Yebra de Basa, Huesca) using the three elements. These tests were divided into two: a first test set (TS1) checked the degree of success of the communication between the base station and the CubeSat at different separation distances; a second test (TS2) set ensured complete base–pilot communication with the CubeSat acting as a repeater.

### 5.1. TS1: Communication Base Station and CubeSat

The base station was located at position 42º31′55.5″ N and 0º11′37.8″ W; four distance values are considered for this location: 186.96 m, 2.1 km, 4.79 km, and 14.20 km. We can observe the values of the received signal strength indicator (RSSI) for these distances in Table 2. In addition, the configurations of the experimentation carried out outdoors for each of the four tests in TS1, ordered by distance from the base station, are collected in Table 2. The RSSI values were below the receptor sensibility of −148 dBm. Hence, the communication was correct. This indicator represents the power level received by the receiving radio after antenna and possible link loss. The intuition behind the RSSI value is that a greater value indicates a stronger signal and, consequently, more robust communication.

**Table 2.** Test results from TS1. The base station is fixed at the location 42º31′55.5″ N and 0º11′37.8″ W. Each test # belongs to a different location of the CubeSat (Latitude/Longitude/Altitude (m)), resulting in a specific distance between these two elements. The output of these tests is expressed in the RSSI value.

| Test # | Latitude | Longitude | Altitude (m) | Distance (km) | RSSI (dBm) |
|:------:|:--------:|:---------:|:------------:|:-------------:|:----------:|
| 1 | 42º31′55.7″ N | 0º22′46.0″ W | 832.98 | 0.187 | −99 |
| 2 | 42º32′26.3″ N | 0º23′59.4″ W | 793.86 | 2.1 | −102 |
| 3 | 42º32′25.2″ N | 0º23′55.8″ W | 823.33 | 4.79 | −103 |
| 4 | 42º35′18.6″ N | 0º31′57.7″ W | 1067.8 | 14.20 | −104 |

*5.2. TS2: Communication between Base Station and Pilot Station Through CubeSat*

Once the first two elements had been validated (base station and CubeSat), confirming the communication between them, the third element was incorporated for testing: the pilot station. Figure 14 shows the communication results between all the elements. The distance between the CubeSat and the base station was 3.02 km, while the pilot station was 2.67 km away, with average RSSI values (averaged from three tests of each link) of 97.67 and 102.33 dBm, respectively. Figure 14a,b demonstrates successful communication from both the base and pilot stations, respectively. From those images, it is possible to confirm a full conversation between the base station and the pilot station via CubeSat. In addition, the RSSI value was below the receptor sensibility of −148 dBm.

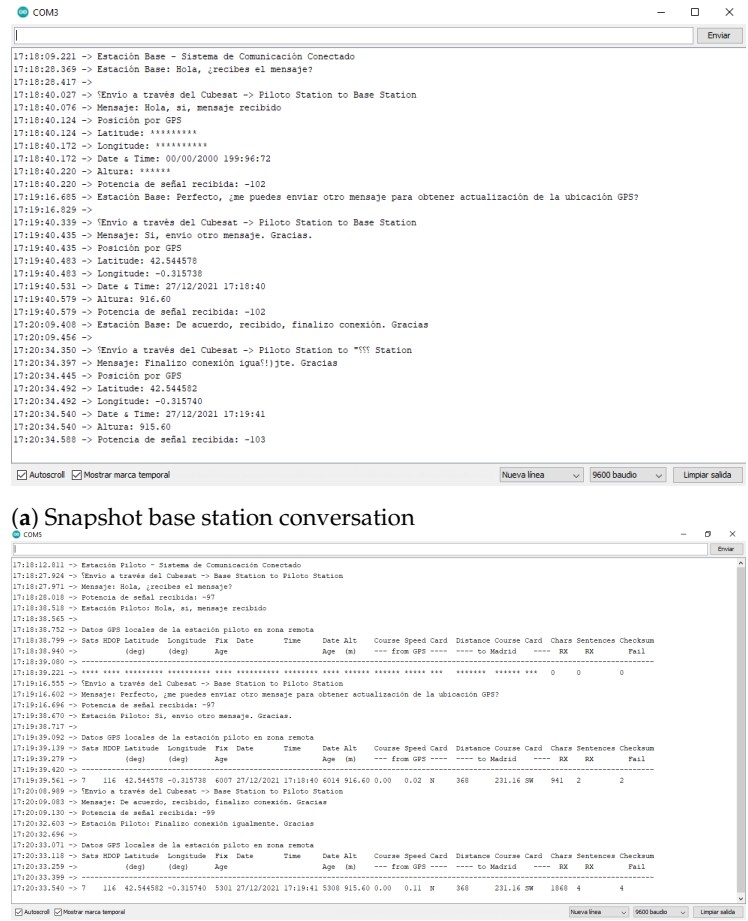

(**a**) Snapshot base station conversation

(**b**) Snapshot pilot station conversation

**Figure 14.** Communication between pilot and base station.

*5.3. Sensor Data Acquisition and Visualization*

As mentioned in Section 3, a dashboard was set up to display the collected data from all embedded sensors within the CubeSat, easily and in real-time. The sensors' values are shown in Figure 15. These results came from the four cases belonging to TS1 between the CubeSat and the base station. Note that the dashboard can be easily modified by either element's results to show their position within the dashboard or the output type.

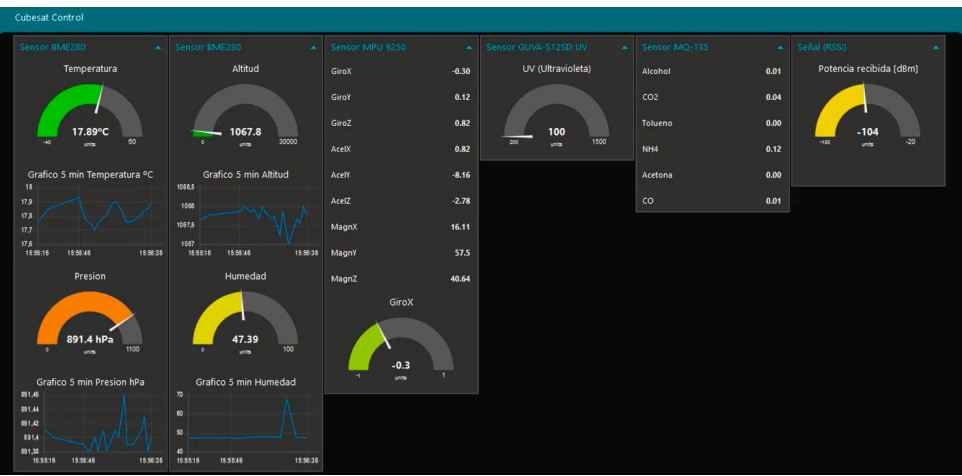

**Figure 15.** A dashboard displays all the CubeSat sensor values in real-time.

Indeed, our system was capable of performing successfully, and end-to-end, pilot-to-base communication was established. Our system can be deployed in less than 2 h using a single 3D printer. The cost of manufacturing and launching is decreased considerably, reducing bureaucracy barriers to using our system openly. Our approach is suitable for remote villages to make them more able to connect via smart devices.

## 6. Conclusions & Future Work

An efficient and fast system for communication is required, due to the lack of communication coverage in 5% of the world and the increment in emergency situations and resulting complex scenarios for remote villages. We proposed a LoRa-based system composed of three devices: a nanosatellite (CubeSat), a pilot station, and a base station. These devices were built using common materials which are available everywhere. Regarding the literature, our solution was a do-it-yourself approach with fast assembling and configuration to be easily deployed in hazard events using emerging communication technologies. The system was tested outdoors, where successful communication between the pilot and base station was performed. The longest distance tested between the two elements was greater than 14 km. Hence, a minimum end-to-end communication link above 28 km can be established quickly, considering the sum of the tested distance of 14 km from the pilot station to the CubeSat and another 14 km from the CubeSat to the base station. In addition, the proposed CubeSat is an in-house solution which is manufactured by a 3D printer, reducing the implementation, deployment time, and launch procedure based on a balloon platform, compared to the literature. In terms of real operation in space, taking into account the use of the proposed microcontroller circuits, 3D-based fabrication, and the suggested earth orbits, the use of the CubeSat is feasible. However, the components' lifetime may be lower than if other CubeSat technologies were involved.

For future work, we plan, but are not limited to, the following:

- Design a printed circuit prototype to house the microcontroller, sensors, and actuators, among others;
- Investigate modification of the control subsystem to be governed by a more efficient and robust microcontroller such as the MSP432;

- Provide more functionality to the nanosatellite: battery level sensor, antenna deployment warning device, and solar panel charge level, among others;
- Design and manufacture another type of antenna to cover larger distances.
- Develop a prototype that can achieve higher heights while maintaining LoRa communication. These heights will be closer to commercial CubeSats or LEO satellites.

**Author Contributions:** Conceptualization, R.P., V.M.B., D.N.B.-I. and C.M.; Investigation, R.P. and D.N.B.-I.; Resources: R.P., V.M.B., D.N.B.-I. and C.M.; Software, R.P. and D.N.B.-I.; Supervision: R.P. and D.N.B.-I.; Validation, R.P., V.M.B., D.N.B.-I. and C.M.; Writing—original draft preparation, R.P., V.M.B., D.N.B.-I. and C.M. All authors have read and agreed to the published version of the manuscript.

**Funding:** This research received no external funding.

**Data Availability Statement:** Not applicable.

**Acknowledgments:** The authors would like to thank Universitat Oberta de Catalunya for its support in the development of this work.

**Conflicts of Interest:** The authors declare no conflict of interest.

## Abbreviations

The following abbreviations are used in this manuscript:

| | |
|---|---|
| 6LoWPAN | IPv6 over Low-Power Wireless Personal Area Networks |
| COTS | Commercial Off-The-Shelf |
| GPS | Global Positioning System |
| GSM | Global System for Mobile Communication |
| HAPS | High-Altitude Platform Stations |
| IDE | Integrated Development Environment |
| IoT | Internet of Things |
| LEO | Low Earth Orbit |
| LoRa | Long Range |
| LoRaWAN | Lomg-Range Wide Area Networks |
| LTE | Long-Term Evolution |
| RSSI | Received Signal Strength Indicator |
| SatCom | Satellite Communication |
| SDR | Software Defined Radio |
| SF | Spreading Factor |
| SPI | Serial Peripheral Interface |
| UMTS | Universal Mobile Telecommunications System |

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
