# Peer review of "LoRa-Based Low-Cost Nanosatellite for Emerging Communication Networks in Complex Scenarios"

_aerospace, doi:10.3390/aerospace10090754_

Round 1
Reviewer 1 Report
The paper presents a SatCom solution for repeating ground base-station signals in cases of problematic coverage. The idea is in line with contemporary SatCom backhauling solutions and is interesting to the potential readers of the journal. The paper is very well structured/written but should be improved so as to address the following issues:
1. The dipole antenna should be adequately described/modeled. Due to the innate assymetry of the dipole-surrounding device shown in Fig. 2, how the azimuth pattern of Fig. 3 is left-right symmetric?
2. What exactly means that the dipole was simulated by means of Matlab? Whic tool/method was used?
3. With regard to the TS1, it is stated that 'The RSSI values are below the receptor sensibility of -148 dBm.' But the values tabulated in the Table of Fig. 4 are obviously above the level of -148 dBm.
4. How can the results of TS1/TS2 can be extrapolated to predict the performance of the system when implemented for the uses mentioned in Section 2 (distances up to 700+km)?
Author Response
Thank you for your thoughtful and insightful comments on our manuscript. We appreciate you taking the time to read our work carefully and provide such detailed feedback. Please, find enclosed our response.

Reviewer 2 Report
First of all, I want to congratulate the authors for their efforts in this manuscript. The topic that they present is interesting and fits within the journal's scope. They have proposed and analysed a LoRa-based Low-cost Nanosatellite for Emerging Communication Networks in Complex Scenarios. There are a few issues to be solved before accepting the paper:
Abstract and keywords:
The abstract should be extended to reach 250 words approx.
The abstract must start with a short description of the problem that the authors are investigating.
In the abstract, the authors have to highlight their results, including numerical values of the performance of their proposal.
Avoid using as a keyword, terms already used in the title. Deleted keywords included in the title and provided new keywords.
Introduction:
There is a lack of references in the introduction to contextualise certain issues. Please provide references to justify the information included in the introduction. For example, the authors have to contextualise the use of LoRa networks in other uses (10.1007/s11036-022-01994-8 and 10.1109/I2CACIS.2019.8825008). At least 10 to 12 references are required in the introduction.
The aim of the paper must be presented in an independent paragraph and extended.
State of the art:
At the beginning of the related work, the authors must add a short paragraph introducing the content.
At the end of the related work section, authors must identify the gap in the current proposals and detail how their proposal will fulfill this gap.
System description:
All the figures must be cited in the text before they appear in the paper.
Please provide the datasheet as a reference for used equipment/devices or include the manufacturer's name and location in the main text (Section 3.1).
A scheme or picture of the proposed CubeSat is required in this subsection. Figure 4 can be moved here, but the dimensions, materials and production stages among others, are also required.
Subsections 3.2 and 3.3 must be extended, highlighting the novelty of the proposal.
Pre-launch procedures:
More information about the launch details is needed, such as location, date, climate requirements, etc…
Test verification and evaluation:
The text of Figure 5 must be translated to English and enlarged to ensure that the image can be read in the paper's printed version.
In the results, the authors have to provide more data than a simple capture of generated information at a precise moment of the launch. The whole generated dataset must be presented in the results. This is particularly critical in the case of RSSI values.
The authors have to justify the selected parameters summarised in Table 3 in the main text.
Conclusion:
The authors have to extend the future work by considering additional modifications or other scenarios.
Author Response

(The authors gave the same response as above.)

Round 2
Reviewer 2 Report
Authors have fixed all my comments. The paper is ready to be published.